# Flood modeling can make a difference: Disaster risk-reduction and resilience-building in urban areas

Jorge A. Ramirez<sup>1</sup>, Umamaheshwaran Rajasekar<sup>2</sup>, Dhruvesh P. Patel<sup>3</sup>, Tom J. Coulthard<sup>4</sup>, Margreth Keiler<sup>1</sup>

<sup>1</sup>University of Bern, Institute of Geography, Bern, Switzerland
 <sup>2</sup>TARU Leading Edge Pvt. Ltd., New Delhi, India
 <sup>3</sup>School of Technology, Pandit Deendayal Petroleum University, Gujarat, India
 <sup>4</sup>School of Environmental Sciences, University of Hull, Hull, United Kingdom

Correspondence to: Jorge A. Ramirez (jorge.ramirez@giub.unibe.ch)

- Abstract. Surat, India is a coastal city with a population of approximately 4.5 million people that lies on the banks of the river Tapi and is located 100 km downstream from the Ukai dam. Given Surat's geographic location the city is repeatedly exposed to flooding caused by large emergency dam releases into the Tapi river combined with high tide water levels. Flood events of this type occur twice a decade, but their frequency and magnitude may increase due to the urbanization, encroachment in flood plain and climate change. A first step towards strengthening resilience in Surat requires a robust method for mapping flood
- exposure at fine spatial resolution. Here, in this study we have developed such a method for Surat using a reduced-complexity hydrodynamic model to simulate flooding, but is easily transferable to other urban locations. Our method features three distinct phases that involve: 1) modelling dam release discharge from the Ukai dam arriving at Surat, 2) modelling flooding within Surat caused by the combination of dam release and tides, and, 3) identifying Surat critical infrastructure, population, and income groups exposed to flooding. Our flood model of Surat utilizes topography produced using elevation data collected from
- an extensive survey. Within the city we have modelled flood scenarios that represent the uncertainty in flood peak discharge and duration resulting from possible climate change. These scenarios include catastrophic conditions that flood 50% of the city and expose > 60% of the population and critical infrastructure to deep flooding. Finally, we highlight how our modelling has contributed to changes in flood risk management within the city following a major flood and resulted in actions that have increased community resilience to flood hazard.

# 25 1 Introduction

Recent efforts in hydrology have highlighted the need to consider the impact that society has on flooding (Baldassarre et al., 2013; Di Baldassarre et al., 2015). Of importance is determining the combination of physical processes and human dynamics that create conditions leading to flood disasters and increased human exposure to flooding. Physical processes can include long-term climatic changes that increase the occurrence of short-term extreme precipitation events triggering flash floods

within rivers. Likewise changes in population, land use, and economics can encourage settlements within floodplains and

subsequently create the need for flood protection measures (e.g. dams, levees, canals) that significantly alter river processes (Wohl et al., 2015). Globally these societal pressures are most apparent in urban areas that are undergoing rapid development (Seto et al., 2011) and an expected 60% increase in population (4 to 6 billion) by 2050 (United Nations, 2015). This projected urban expansion will mostly occur in Asia, specifically in locations with dense populations and low topography susceptible to

5 flooding (Güneralp et al., 2015; Jongman et al., 2012). For these reasons there is an urgent need to estimate flood exposure in Asian urban cities as well as translate these estimates into actions to reduce exposure and disaster risk to urban inhabitants and critical infrastructure.

One method to estimate flood exposure are computer models that generate predictions of flood inundation that can be spatially overlaid on maps of population, critical infrastructure, and income groups. For example, recent advances in reducedcomplexity flood models (Bates et al., 2010; Coulthard et al., 2013a) with simplified physics have shown promise in estimating flood depths, extent and velocity in urban areas (Fewtrell et al., 2011; Ramirez et al., 2016; Sampson et al., 2014). These models are particularly suited for data sparse locations, with large spatial extents (> 500 km<sup>2</sup>), and urban topography represented by fine spatial resolution ( $\leq$  30 m) digital elevation models (DEMs). Moreover, open source software and the low

- computational overhead needed for such reduced-complexity modelling facilitates the generation of multiple model runs that produce flood maps representing ensemble simulations of various flood scenarios. As such, flood exposure estimated with reduced-complexity models can explore uncertainty stemming from environmental forcings that result in various flood return periods and duration. Herein we use a freely available, reduced-complexity flood model to estimate flood exposure for the first time in a large, densely populated city in northwest India where human intervention (e.g. dam and levee construction) and
- demands on the hydrological cycle (e.g. water for irrigation) has contributed towards multiple catastrophic floods. Like many urban areas, flood mapping and exposure assessment in our study site is challenged by the scarcity of available data, in this paper methods to gather such data and forecast flood inundation using open-source software are presented. An important part of this research is how results from our model have directly motivated changes in the way local authorities are preventing future floods and increasing the resilience of the city.

# 25 2 Data and methods

## 2.1 Study site

In India, the National Disaster Management Authority has provided guidelines that specify that all cities prone to flood have a detailed flood preparedness plan prepared at the ward level (city administrative unit). These efforts are intended to "minimise vulnerability to floods and consequent loss of lives, livelihood systems, property and damage to infrastructure and public

utilities" (National Disaster Management Authority, 2008). Recent enforcement has led cities to compile a database of resource inventory which includes tools and equipment needed for response post flooding. Surat, is one of these cities, and is located in the northwest of India near the Arabian Sea and is a major port, industrial, and commercial center (Fig. 1c). Surat is the seventh

largest city in India with an area of 325,000 km<sup>2</sup> and a population of 4.5 million. From 1990-2014 the city underwent considerable urban development within the Tapi river floodplain that resulted in a 170% increase in urban land cover (Misra and Balaji, 2015). Upstream from Surat the Tapi river has an approximate length of 634 km up to the origin and a catchment area of 62,000 km<sup>2</sup> (Fig. 1a). The catchment lies within a highly variable rainfall zone where 90% of the annual rainfall is

- received during the monsoon months (June to October) and an average annual rainfall of 830 mm (Kale and Hire, 2004). In order to address the problem of flooding in Surat, storing water for summer irrigational needs, and hydropower the government of India constructed the Ukai dam on the river Tapi in 1971 (Fig. 1b). The multi-purpose nature of the dam forces the irrigation authorities to conserve water to the maximum allowable capacity of the reservoir during the monsoon. Thus, any extreme rainfall experienced at the end of the monsoon leads to unplanned releases of water from the reservoir that may cause flooding
- 100 km downstream within Surat. As such, since the construction of the dam floods in Surat are socio-natural hazards (Birkmann et al., 2013) because flood events are caused by the combination of natural phenomena (e.g. rainfall) and manmade causes (e.g. dam mismanagement). Over the past half century the city has experienced on average two floods per decade that were caused by dam releases. This includes catastrophic flood in the years of 1994, 1998 and 2006. The most recent flood in August 2006 was the result of intense rainfall in the Tapi catchment that led to a 15 day dam release with a peak discharge
- near 25,000 m<sup>3</sup> s<sup>-1</sup> (avg. discharge= 9,300 m<sup>3</sup> s<sup>-1</sup>, low flow = 640 m<sup>3</sup> s<sup>-1</sup>). During this catastrophic flood 75% of the city was flooded, an estimated 40% of the city inhabitants received no warning about the dam release, human deaths totalled 300 and damages exceeded 3 billion USD (Bhat et al., 2013; Patel and Srivastava, 2013). Since 2006 embankments alongside the river have been raised and no major floods have occurred, but the effectiveness of these embankments is questionable because future climate change within the Tapi basin during the monsoon season is expected to have a 30% increase in rainfall (Mishra and
- Lilhare, 2016) and increased occurrence of extreme rainfall events (200-350 mm/day) (Bhat et al., 2013; Deshpande et al., 2016). These climatic changes may result in a 75% increase stream flow for the Tapi river (Mishra and Lilhare, 2016) and subsequently result in unprecedented emergency dam releases that produce flooding in Surat. Although climate change will be a major driver of future flooding in Surat, likewise urban flood risk will also be strongly dependent on the increase of urban expansion in the city stemming from socio-economic changes (Güneralp et al., 2015). Foreseeing future flood disasters,
- activities have been developed by the Surat Municipal Corporation and a commissioner that supervises flood management activities with the support from ward level engineers. One effort enacted in Surat to prepare for flooding was the production of flood maps that did not previously exist for the city wards. In the next section we present flood simulations that were commissioned to understand the extent and magnitude of impact of different quantity of water released from the dam.

#### 2.2 Modelling

Our modelling of flood in Surat featured two distinct hydrodynamic modelling stages. In stage one, we modelled dam release from the Ukai dam and the flow of water within the Tapi to the eastern boundary of Surat. Afterwards, the second stage of modelling used the outputs from stage one to drive a flood model of Surat that also included tidal effects on the river. In both modelling stages CAESAR-Lisflood models were used. CAESAR-Lisflood model is the outcome of merging the CAESAR

landscape evolution model (Coulthard et al., 2013a) with the latest model version of Lisflood-FP (Bates et al., 2010). CAESAR-Lisflood is a storage cell model, where a DEM represents the landscape and water is stored at the raster cell locations. Water is routed over the landscape in two dimensions (2D) from raster cell to cell using a simplification of the shallow water equations (Bates et al., 2010). CAESAR-Lisflood is open source, has low computational demands, low data requirements and has been used to replicate fluvial (Coulthard et al., 2013a, 2013b) and storm surge (Ramirez et al., 2016; Skinner et al., 2015) flooding. The model operates on inexpensive PC hardware, is parallelised to utilise multiple processor cores and is driven by a GUI front end making model set up and operation simpler. These model characterises make CAESAR-Lisflood ideally suited for exploring the uncertainty in dam releases resulting from future climatic changes within the Tapi basin. We have developed dam release scenarios with different flood peak discharge and duration and estimate the resulting

flood exposure within Surat.

For the first stage of modelling we developed a CAESAR-Lisflood hydrodynamic model at a spatial resolution of 30 m cell size (2.2 million raster cells), of the Tapi river from the Ukai dam to Surat (Fig. 1b). This DEM contains 79 km of the river length and nearly 15 km of floodplain on either side of the river. The hydrodynamic model uses topographic information

- collected by the Shuttle Radar Topography Mission (SRTM) between February 11-22, 2000 (Jarvis et al., 2008). During this time the Tapi river was under low flow conditions (mean discharge =  $129 \text{ m}^3 \text{ s}^{-1}$ ) and the imaged river reach elevation values downstream from the Kakrapar weir sufficiently characterized the river channel (Fig. 1b). River channel elevation values upstream from the Kakrapar weir to the Ukai dam contained the elevation of the water level behind the weir. To better represent the river bed these elevations were replaced with values that were calculated using the bed slope (m = 0.0006) of the river
- downstream from the weir. Vegetation heights in the SRTM DEM can affect flood model results (Baugh et al., 2013; Wilson et al., 2007) but a 'Bare-Earth' SRTM DEM at 30 m resolution does not exist. Using a 90 m spatial resolution 'Bare-Earth' SRTM DEM (O'Loughlin et al., 2016) and the original 90 m SRTM DEM we estimated the mean vegetation height of our DEM was 1.3 m. We concluded that vegetation effects would be minimal at our site and chose not to correct the DEM for these artefacts.

The flow of water between raster cells in CAESAR-Lisflood requires a roughness coefficient per cell (Manning's n) to represent landscape resistance. Throughout the river floodplain a 30 m spatial resolution land cover map (Yu et al., 2013) was acquired and roughness coefficients were applied to each land use class using values reported in Alfieri et al. (2014). Channel roughness was determined by calibrating the model using the September 22-26, 2013 Ukai dam release (Fig. 2a) that produced

minimal amounts of flooding in Surat. For the calibration the modelled reach was extended 22 km downstream to the nearest gauging station which is not affected by tides (Singanpur weir (Fig. 1c)). Channel roughness values representing the sandy bed of the Tapi (Kale and Hire, 2004) were trialled (Manning's n = 0.01, 0.02, 0.03 (Fisher and Dawson, 2003)) and the observed hydrograph recorded at the weir was reasonably replicated by the model using a channel roughness of 0.02 (Fig. 2a).

Using this channel roughness the root mean squared error between observed and modelled water levels at the weir was 0.39 m.

- Synthetic Ukai dam outflows were based on the outflow of August 4-12, 2006 that produced catastrophic flooding in Surat. A
  nonlinear regression was performed on this dam outflow time series to produce an idealized dam outflow at 10 min time steps (Fig. 2b). Multiplication factors were applied to the idealized dam outflow to produce synthetic dam outflows with all combinations of outflow durations (2, 3, 4, 5, 6 days) and estimated return periods (5, 10, 25, 50, 100, 250 year) for the Tapi river (Rakhecha and Clark, 2002) (Fig. 2b). A total of 30 dam outflow scenarios were produced with each scenario consisting of two days of base flow (2000 m<sup>3</sup> s<sup>-1</sup>) to develop initial hydraulic conditions, followed by a period of dam release, and
- concluding with two days of base flow. For each scenario discharge arriving near Surat (western edge of the DEM) was collected every 10 minutes to produce a hydrograph that served as input into the second stage of modelling.

For the second stage of modelling a separate CAESAR-Lisflood hydrodynamic model was developed to estimate flooding within Surat. An accurate flood assessment of Surat required a representation of the bare earth that did not include the elevation

- of urban infrastructure and vegetation heights. A high resolution DEM (e.g LIDAR quality) did not exist for the city so topographic data of the city was produced by undertaking an elevation survey of Surat using a differential global positioning system (DGPS). The city was intensively surveyed to acquire point elevations of hydrologically important topographic features that included streets, depressions, peaks, and river embankments. In total 6,219 point elevations were acquired during this ground survey with a horizontal accuracy of  $\pm$  30 cm and vertical accuracy of  $\pm$  30 cm. Additionally, 172 river bed cross-
- sections were obtained from the irrigation department to represent the river bed elevations. Outside of the city limits, where the survey was not preformed, elevation information was obtained from a 30 m SRTM DEM and converted into point vector GIS (Geographic Information System) format. Using DigitalGlobe high-resolution imagery (30 cm pixel size) locations surveyed consisting of bare ground or roads were used to determine the elevation difference between SRTM and survey data locations (n = 48). This vertical offset was applied to the survey data and transformed all the survey elevation values into the
- SRTM vertical datum (EGM96). Survey points, river cross-sections and the surrounding SRTM values were interpolated together using the ordinary kriging method (ESRI ArcMap 9.3) to produce a high resolution DEM (30 m) of Surat, surrounding peri-urban areas and the Tapi river bed (~1 million raster cells) (Fig. 1c). Overall this DEM encompasses an area of 835 km<sup>2</sup> and contains 45 km of the Tapi river commencing near the eastern boundary of Surat and ending in the Arabian Sea. Channel roughness of the first modelling stage was also used in the hydrodynamic models of Surat (n = 0.02). We chose to represent
- Surat's urban-fabric and the peri-urban areas with roughness values based on land cover classes (Alfieri et al., 2014) from a 30 m spatial resolution land cover map (Yu et al., 2013). Ideally we would have calibrated the roughness of the city and floodplain by replicating an observed flood extent, but this data does not exist for the DEM we have developed because embankments within the city have been raised and a major flood has not occurred since the construction of these embankments.

Flood models of Surat were driven by 10 min temporal resolution discharge from the output from the first stage of modelling and this discharge was added to the upstream end of the Surat reach (Fig. 1c). Additionally, the effect of tides on the river was simultaneously modelled in all Surat flood simulations. Tidal fluctuations at Surat were modelled using the WebTide tidal prediction model (Dupont et al., 2002) that produces tidal heights for a particular time period using tidal harmonic constants

- of major tidal constituents pre-calculated from a tidal ocean model. Tidal predictions of 5 min temporal resolution for Surat were calculated with WebTide for the years between 2010-2015 and monsoon months when emergency dam releases are more likely to occur (August, September and October). From this time series a subset spanning 240 hrs (10 days) was extracted containing the maximum high tide of 3.1 m and a maximum tidal range of 5.9 m. Tidal fluctuations were reproduced in the model by gradually raising and lowering water levels at two nearshore locations west of Surat (Fig. 1c) and this produced a
- tide that advanced upstream the lower portion of the Tapi river. For each flood model maximum high tide coincided with peak discharge arriving at Surat and the combination of these factors produced worst-case flood conditions for the city. Each Surat flood model was set up using the same parameterization and tidal inputs, but the upstream dam release inputs were different. Model output per flood scenario consisted of a map of maximum flood water depths at each DEM location and this map also delineated the maximum flood extent for each flood scenario. Preliminary model flood maps were approved by municipality
- flood experts, and this served as qualitative validation of the flood model extents and water depths.

In our study flood exposure was defined as the extent to which critical infrastructure, people, and income groups are geographically located in areas prone to deep flooding ( $\geq 0.75$  m). We chose to analyse flood exposure at these depths because they result in extreme conditions that are threatening to life (waist deep water) and property (ground floor inundation) (World

- Bank, 2011). Geographically our analysis of flood exposure included Surat and locations beyond the city limits. Herein we made a distinction between critical infrastructure that is related to transport and rescue/intervention and both datasets were analysed separately. Main roads and railroads comprised transport infrastructure, whilst fire stations, police stations, hospitals, and water tanks were regarded as rescue/intervention infrastructure (Fig. 3a). Locations of rescue/intervention infrastructure was mapped through questionnaire based field surveys using a global positioning system, whilst transport infrastructure was
- obtained from existing maps. Exposure to people was made using a map of spatially distributed population estimates from 2015 (Fig. 3b) at the spatial resolution of 100 m<sup>2</sup> (Gaughan et al., 2013). Lastly, we investigated the relationship between flood exposure and monetary income using a map of neighbourhoods with different income levels (Fig. 3c). Income groups were mapped through a detailed physical survey which was conducted over 100 transects across the city and questionnaire based survey conducted across 400 households along these transects to capture their capacity and vulnerability information. A geo-
- location and building typology based association technique was used to spatially identify areas which are similar to the 100 transects based on their proximity to the surveyed sample and physical characteristics. Land use polygons within the neighbourhood having similar characteristics were assigned similar rankings. Flood exposure analysis was performed using the modelled Surat maximum water depth maps resulting from each of the 30 dam release scenarios with different return

periods and flood duration. These flood maps were spatially overlaid on the maps of critical infrastructure, population and income group, and the percentage of each metric at each location with inundation  $\geq 0.75$  m was calculated.

A disaster management cycle for flood hazard summarizes processes and actions taken by government, businesses and the community to reduce the consequences of flooding. Disaster management cycles include phases for mitigation, preparedness, response, and recovery (Keiler et al., 2010; Vasilescu et al., 2008). Mitigation minimizes the effect of flooding and may include raising community awareness about floods or constructing flood protection measures. Preparedness involves the development of plans to respond to flooding and may include the implementation and operation of a flood early warning system. The response phase encompasses actions taken after a flood event to reduce the effects of flooding on the community and consists

10 of rescue efforts and the delivery of aid. A recovery phase is an extended period after the event where the community rebuilds. We use this disaster management cycle to identify phases where our modelling has made tangible contributions to flood risk reduction and resilience building after the 2006 Surat flood.

## **3 Results**

Figure 4 are dam outflows and the resulting city flood hydrographs produced by the dam outflow model (model stage 1). In all of the simulations a quantity of the dam outflow was lost to the topographically flat, expansive, floodplain between the Ukai dam and Surat and this attenuated/distorted the rising limb of the resulting flood hydrograph. Overall this section of river acted as a buffer or store that reduced the amount of water arriving in the city by 1-4%. The buffering capacity of the floodplain was greatest when dam out flows were high and of long duration, and these conditions resulted in extensive amounts of flooding on either side of the river. An important result from the modelled dam outflow was the duration of time required for the flood model to crest the embankments after a dam release. Embankments in the city were constructed to contain floods of 17,000 m<sup>3</sup> s<sup>-1</sup> and

- it was assumed that discharge greater than this would produce flooding within the city. Using this flood threshold, on average all modelled dam releases flooded the city between 10-14 hrs (average = 11 hrs). This information provides the minimum amount of flood warning that can be given to the city inhabitants after a dam release has been performed. Furthermore, all model runs indicated that after maximum dam outflow between 14-33 hrs (average = 26 hrs) elapsed before maximal discharge
- 25 conditions occurred near Surat. This time lag was greatest for dam release scenarios with combined long durations and high peak discharges.

Figure 5 are maps produced by the city flood model (model stage 2) and represent maximum flood depths and spatial extents for a subset of the flood scenarios. Here it can be determined that changes in flood return period (Fig. 5 a,d,g) caused greater
amounts of flooding (extent and depths) than increases in flood duration (Fig. 5 g,h,i). It is apparent that the city levees are effective at preventing flooding at the lowest return period (1:5 yr), but limited amounts of flooding still occurred in the west part of the city (Fig. 5 g,h,i). Levees are less effective at containing floodwaters at return periods ≥ 50 yrs, with extensive

amounts of flooding in the city and peri-urban area. The worst case scenario of a 1:250 yr flood lasting six days resulted in 50% of the study being flooded (Fig. 5c) and nearly all of the flood depths were considered deep ( $\geq 0.75$  m). Large flood extents and deep water depths do not always translate in to high levels of exposure unless flooding spatially coincides with critical infrastructure and residential locations. Figure 6 illustrates that an overall increase in flood return period results in

- greater exposure than increases in flood duration. For each sector analysed exposure continues to be elevated (28-48%) for a flood equivalent to the catastrophic 2006 flood event (1:25 yr return period, 5 day duration), regardless of the construction of new embankments. Specifically, reoccurrence of the 2006 event would flood (water depths ≥ 0.75 m) 29% of the city's population (~713,000 people), 29% of the transportation network, and 37% of the hospitals, police/fire stations, and water tanks. Rare, but plausible, extreme events (1:250 yr return period, flood duration of 6 days) shockingly inundated 60% of the
- population (~1.5 million people), 60% of the rescue/intervention, and 60% of the transportation network. Unexpectedly, the newly constructed flood protection does not eliminate flooding and exposure from low return period events (1:5 yr, 2-4 days duration). For example, the smallest flood modelled (1:5 yr, 2 day duration) can potentially expose ~ 100,000 people to deep flooding. Regarding exposure and income level (Fig. 6 d,e,f), model results suggested that low income communities are less exposed than, moderate and high income communities.

Using the disaster management cycle (mitigation, preparedness, response, and recovery) our flood maps and exposure assessment contributed to several phases reducing flood risk reduction in Surat. Within the *mitigation* phase our flood maps and exposure assessment have contributed to the reduction of the impact of flooding by instigating changes within the operational protocols of the urban local body and attitudinal change amongst the citizens especially the vulnerable population.

- The floods maps are currently being used to strengthen the city disaster management plan and resource allocation during monsoon preparedness. The results from the flood model were also used for demarcating possible flood depths on public lamp posts across the city (Fig. 7). The demarcation raised awareness of flooding by providing information to people on the extent of inundation for specific quantities and durations of water released from the dam. This flood depth information is particularly important to Surat's immigrants, which comprises the majority of the population. Between 2001 and 2011 the city's population
- increased by over 70% (India, 2011) due to pull migration, where people come to the city from both within the state of Gujarat and from other states such as Bihar and Odisha in search of employment. Many of these immigrants who come to the city choose residences in areas which are more vulnerable to flooding. Such choice is usually influenced by the cost and rarely is information on flood levels at these locations available. The painting of lamp posts throughout the city with potential flood heights from the model provides dual benefits to the citizens. Firstly, migrants and visitors to the city will be aware of areas
- which are prone to the floods. Thereby they will make informed decisions while choosing their place of residence. Since the implementation of the painted lamp posts this has already started impacting the real estate within the city. Many people now are re-considering their choice to invest in buildings which are highly vulnerable to the floods. Secondly, the marking also act as communication posts during an event. The markings help people internalize the technical information which are disseminated during the event by the different government agencies. For example, if a dam release occurs for a specific return

period, any resident can look at the lamp post near their residence and get information on the extent of inundation which is likely to occur in the surroundings. Contextualized information provided to different communities across the city will not only help citizens prepare for the flood hazard but will also help others in low risk zones to remain calm. This leads to less city wide panic, reduced traffic congestion and reduced social unrest before the event. In addition to the painted lamp posts, flood maps produced by the model are currently being used by Municipal Corporation engineers and their ward level representatives for

produced by the model are currently being used by Municipal Corporation engineers and their ward level representatives for their monsoon preparedness activities which includes public awareness program on flood risk reduction.

Within the *preparedness* phase flood maps generated from the model are being used for preparing flood risk management plans that include: 1) highlighting key locations for helicopters to drop food and medical supplies, 2) locate health care units

- which may require evacuation in case of flooding, 3) identify buildings which are to be used as safe shelters, 4) determine critical infrastructure that needs to be protected and 5) more importantly identification of vulnerable locations where there will be a need for prioritized communication for evacuation. The flood maps are currently being used as a part of a larger End to Early warning system which was established under the Asian Cities Climate Change Resilience Network (ACCCRN) and currently being maintained by the Surat Municipal Corporation and Surat Climate Change Trust. Shortly after a dam release,
- within the *response* phase, the exposure information will provide city officials with an overall estimate of population and infrastructure that are likely to be affected by flooding. The depth of the flooding as modeled using multiple simulations will help them in prioritizing their response actions. Usually high priority is given to slums (informal and formal settlements) where the water depth will be more than 2 m. Also all other settlements including middle income and high income communities are informed through public address system to initiate evacuation. Previously, there used be a city wide announcement of warning
- and alerts followed by deployment of resources across all areas based on the administrative boundaries. With the help of the flood maps, the city government can provide contextual warning and allocate their resources on select areas depending upon the extent of the possible impact. Additionally modelled flood maps will be used to locate communities that require immediate evacuation. During the *recovery* phase model output is not useful, instead resource allocation usually happens post field investigation of actual damage.

#### 25 4 Discussion

Our modelling approach used a 2D hydrodynamic model to estimate the quantity and velocity of water arriving at Surat and represents an improvement over previous studies investigating flooding in Surat using one dimensional (1D) approaches (Timbadiya et al., 2014a, 2014b). While 1D models can adequately predict water levels that are confined to a river channel, a 1D model's performance deteriorates when water spreads onto the floodplain and flow becomes 2D. As such, our 2D implementation provides an advantage over 1D models by better replicating the important process whereby flood wave attenuation, loss of water, occurs from the dam to the city when water spreads onto the floodplain (Fig. 4). Further improvements can be made to our models by including processes that replicate infiltration to the floodplain subsurface. In our

simulations flood discharge conveyed across the floodplain accounted for a loss of 1-4% of the water arriving in Surat, but these values are more than likely conservative losses of water because infiltration was not modelled. As such greater flood attenuation and less flooding in Surat could be attained with a model that accounts for infiltration occurring on the floodplain. Regarding flow velocity, our model was able to replicate well the amount of time it takes for flooding to commence in Surat after a major dam release (> 17,000 m<sup>3</sup> s<sup>-1</sup>). For example, during the 2006 flood the dam release flood wave took approximately 10 hrs to arrive in Surat, and our model estimated 12 hrs for a dam release of similar peak discharge and duration.

Uncertainty in the flood maps produced by CAESAR-Lisflood could be traced back to error in data sources and model processes. The principal source of uncertainty in CEASAR-Lisflood's output is produced by error in the topographic data that

- subsequently affects the flood wave movement. For example the smallest elevation error, obstruction, or artefact in the DEM could divert water to locations that do not experience flooding. In our city flood model topographic error was minimized by not using a DEM that contained land features (buildings and vegetation), but instead, a bare surface representation of the city was developed from a high resolution ground survey. We find that this method of generating a bare surface DEM is effective for urban environments where LIDAR quality topographic data is not available or feasible. Although the DEM created is of
- superior quality (vertical and horizontal accuracy) than the commonly used 30 m SRTM DEM, more extensive surveying is needed to achieve greater spatial resolution (< 10 m). Of particular importance is the need to accurately represent the city embankments within the DEM and model output suggests that embankments were adequately represented in the DEM because all city models do not commence breaching embankments until discharge exceeds the established flood warning level for the city (~ 17,000 m<sup>3</sup> s<sup>-1</sup>).

A secondary factor producing uncertainty in the city model output is error in surface friction values that were used to represent the restriction of flow over land features. Although the finest spatial resolution (30 m) land cover data available was used to infer surface friction values for Surat, it is possible that classification error in the land cover resulted in incorrect setting of some surface friction values. Ideally channel roughness for this model would be calibrated, but observed flood extents and water depths do not exist to calibrate the floodplain friction for the city. These data are not available because no major flood has occurred since the construction of the embankments. Regardless, in order to partially account for the uncertainty in floodplain friction and resulting spatial extent of flooding we present 30 possible dam relea