# Peer review of "Flood modeling can make a difference: Disaster risk-reduction and resilience-building in urban areas"

_Hydrology and Earth System Sciences, 2016_

## Referee Comment (RC1) · Anonymous Referee #1 · 14 Dec 2016

Urban flood modelling in Surat is performed by using a standard flood model and widely used topographic data. The study is overall technically sound, but some choices (e.g. parameterisation strategy) are not justified. Also, previous scientific work on calibration, validation and uncertainty analysis (e.g. effective roughness coefficients) in flood modelling (chichis huge) has been not recognised. Nor did the paper refer to the tons of research papers that have assess the pros and cons in using SRTM data to build hydraulic models. For this reason, the paper did not convince me about the novelty of this study. In fact, I don't see any progress beyond the state of the art. It looks to me as a standard model application. Link to disaster management cycle could have been potentially interested, but has not been sufficiently developed. What is stated

with this respect can be said for any modelling exercise around the world. The introduction also mentions that an "important part of this research... motivated changes in the way local authorities...". This has triggered my curiosity, but I could not find any scientifically-sound work about the way in which flood modelling leads to e.g. changes in risk reduction policy. In summary, this paper does not provide any substantial new concepts, ideas, methods, or data, and I do not recommend its publication.

---

## Author Comment (AC1) · 19 Dec 2016

To respond to the reviewer, we believe our manuscript is novel for several reasons. Crucially, we address the challenges of modelling flood exposure in data scarce, densely populated, urban regions by developing a method that utilizes an open source model and freely available data sets.

- Our modelling approach is computationally efficient and can be applied over large spatial and temporal extent without sacrificing fine-scale detail in topography (30 m).

- Within this method we consider uncertainty in flooding by adopting a scenario based approach to estimate flood depths and exposure for critical infrastructure, people, and

income groups.

- Our results produced the first flood maps for a large Indian city (Surat) that explores the uncertainty in flood peak discharge and duration resulting from possible climate change and also estimate how this uncertainty cascades to flood exposure. These scenarios include catastrophic conditions that demonstrate the ineffectiveness of human alterations to the landscape (e.g. embankments) to protect the city from flooding.

Further novelty in our research is demonstrating how modelling directly motivated changes in preventing future floods and increasing the resilience of the city. These changes include: 1) demarcating flood depths on public lamp posts across the city, 2) preparing flood risk management plans with the flood maps (e.g. key locations to drop food and medical supplies) and 3) prioritizing the evacuation of the city using the flood maps. The results of this exercise is currently being used by the local government as a part of India's first urban end to end early warning system for floods. This work is currently acting as an exemplar motivating other Indian cities to undertake similar modeling exercises. Throughout our manuscript we have mentioned these points and provide additional information.
* * *

---

## Referee Comment (RC2) · Anonymous Referee #2 · 5 Feb 2017

Journal: HESS
Title: Flood modeling can make a difference: Disaster risk-reduction and resilience-building in urban areas
Author(s): Jorge A. Ramirez et al.
MS No.: hess-2016-544

**General comments**

The manuscript "Flood modeling can make a difference: Disaster risk-reduction and resilience-building in urban areas" by Jorge A. Ramirez, Umamaheshwaran Rajasekar, Dhruvesh P. Patel, Tom J. Coulthard and Margreth Keiler demonstrates a simple approach for flood water depth modeling over the city of Surat, India, by simulating dam-release scenarios with different flood peak discharge (return periods) and different flood durations. The authors introduce some refinements of the exiting modeling tool (resolution at 30 m, removal of water level behind the weir, base earth condition, and surface roughness), which was not the main purpose of the work though.

The main objective of the work is, using the model, 1) to estimate and interpret the possible flood exposures with respect to factors such as critical infrastructure, population, and income groups, and 2) to find the contribution of the modeling results within the four-phases (mitigation, preparedness, response, and recovery) of the disaster management cycle.

The motivation, objective and the tools are very interesting and suitable for the topics of the HESS. However, the simulation-strategy of the model is poorly designed with insufficient evidences, which hampers in-depth analysis of the results (see also the below major and minor comments). Therefore, I would not recommend publication of this manuscript in the HESS.

Recommendation: Reject.

**Major comments**

1. Poor analyses and lack of supporting materials (Results, Section 3)
The results are shown in two parts:
   **i) The flood exposure** with location of critical infrastructure (1-transport, 2-fire/police/hospitals/water tanks), population, income groups that are prone to deep flooding (> 0.75 m). The analyses based on the presented simulations lack in explaining the impact of water release from the dam that is the objective of the work. Also, some statements are not well justified:
   • Page 8 first paragraphs: lines 4-5 "Figure 6 illustrates that an overall increase in flood return period results in greater exposure than increase in flood duration…" I disagree with the interpretation of the authors. The choice of flood duration (2 to 6 days) should be better explained, and further interpretation about how this relates to flood exposure to high return periods should be provided.

**ii) In the context of the disaster management cycle.**
- Pages 8 and 9: "our flood maps and exposure assessment have contributed to the reduction of the impact of flooding by instigating changes within the operational protocols of the urban local body and attitudinal change amongst the citizens…" Except for the changes introduced to the public lamp posts (to identify possible water depth - page 8, line 21), it is not clear how the flood model has actually triggered the mentioned changes. It would be really interesting to provide further description of how the obtained results triggered the mentioned changes. So far, the sentence seems speculative and the supporting material is insufficient, which reduces significantly the interest of the presented work.

2. Discussion

The authors address some general factors that may cause uncertainty in the flood model used. However, it is confusing that some of the conclusions are not logically connected with the results presented in the previous section. For example,
- Page 10, lines 26-27: Although the authors designed 30 possible dam release flood scenarios to account for the uncertainty in flood friction and resulting spatial extent of flooding, the analysis of the results lack in showing how such ensemble could account for what kind of uncertainty.
- Page 11, lines 7-10: The explanations of Fig 6 regarding income groups are not clear.
- The authors describe some future aspects for the use of the obtained flooding maps (Page 11 2$^{nd}$ paragraph) and application done in the past for risk management in the Surat community (Page 12, 1$^{st}$ paragraph). However, the statements are questionable upon the results presented in the manuscript. Supporting materials are necessary.

**Minor comments**
- Citation of references, Page 1, 12: Baldassarre et al. 2013 → Di Baldassarre et al. 2013.
- Page 3, line 10: "since the construction of the dam floods in Surat…" Please, revise.
- Page 3, line 24: Typo in the Reference List for (Güneralp et al. 2015)
- Page 4. Line 7: "characterises" → Should not it be "characteristics"?
- Page 7, line 1-2: An example of "these flood maps…was calculated." would be useful to illustrate this concept. Fig 5 could be cited here.
- Fig 2. The indication of the simulation results with respect to the return periods (the ticks on the right Y-axis) is not clear.
- Fig 4 and page 7, lines 20-25: It is unclear how the text relates to what can be seen in the figure. I suggest adding some extra elements in the figure that can help understanding the periods the city was flooded.
- Page 7, line 30-32: "It is apparent that the city levees are effective at…." It is not clear what the authors want to deliver in this sentence and how they see the effectiveness of levees in Fig 5.